# Acceptability of HPV Vaccination in Young Students by Exploring Health Belief Model and Health Literacy

**DOI:** 10.3390/vaccines10070998

**Published:** 2022-06-22

**Authors:** Alessandra Fallucca, Palmira Immordino, Luca Riggio, Alessandra Casuccio, Francesco Vitale, Vincenzo Restivo

**Affiliations:** Department of Health Promotion, Mother and Child Care, Internal Medicine and Medical Specialties, University of Palermo, 90127 Palermo, Italy; alessandra.fallucca@unipa.it (A.F.); palmira.immordino@unipa.it (P.I.); luca.riggio@unipa.it (L.R.); alessandra.casuccio@unipa.it (A.C.); francesco.vitale@unipa.it (F.V.)

**Keywords:** HPV vaccine, behavioral model, Health Belief Model, health literacy, vaccine sources of information, low vaccination coverage

## Abstract

Evidence on the human papillomavirus (HPV) vaccine shows that it is effective in reducing the burden of HPV-related diseases. For more than 15 years the HPV vaccine has been offered free of charge in Italy to girls from the age of 12. Over time, the free offer of the HPV vaccine has also been extended to boys and to young adults at risk of developing HPV lesions. Despite the HPV vaccine’s effectiveness and availability, vaccination coverage is low in Italy, with a reported value of 46.5% in 2020. Furthermore, in the southern administrative regions, vaccination coverage is even lower than national values, with 25.9% coverage in Sicily. A cross-sectional study was conducted among university and high school students in the Palermo area (Sicily, Italy) in order to identify the determinants of HPV vaccination adherence by using a questionnaire that investigated factors of HPV vaccine practice. The study explored the behavioral attitude by using the Health Belief Model (HBM), and also used the SILS test and the METER test to investigate the level of health literacy (HL). Overall, 3,073 students were enrolled, and less than a third reported they had completed the vaccination schedule (*n* = 925, 30.1%). Multivariable analysis showed that the factors directly associated with the adherence to HPV vaccination were female sex (OR = 4.43, *p* < 0.001), high HBM total score (OR = 4.23, *p* < 0.001), good HL level (OR = 1.26, *p* = 0.047), parents (OR = 1.78, *p* = 0.004), general practitioner (OR = 1.88, *p* = 0.001), and educational material provided by public vaccination services (OR = 1.97, *p* = 0.001) as HPV vaccine information sources. Further health-promotion programs focused on improving HL and perception of the HPV vaccine’s benefits should be implemented in order to achieve the desirable 95% vaccination coverage.

## 1. Introduction

Human papillomavirus (HPV) infection is the commonest sexually transmitted viral infection worldwide; this small virus is highly common and widespread, with a marked tropism for skin and mucous membranes [1]. There are more than 200 types of HPV that can be classified in different groups according to the level of oncogenic risk [2]. HPV is a well-established cause of cervix, anogenital region (anus, vulva, vagina, and penis), and head and neck cancers, as well as genital warts, in both sexes [3]. Worldwide, cervical cancer is by far the most common HPV-related neoplasm, and it represents the fourth type of cancer both by incidence and mortality, with more than 560,000 new cases and more than 310,000 deaths per year [4]. Nearly all cases of cervical cancer can be attributed to HPV infection. There is no specific treatment to counteract HPV infection, and prevention is the most effective strategy, including both immunization and cervical cancer screening. The elimination of cervical cancer is today a global public health goal launched by the WHO in 2018, with the aim to vaccinate at least 90% of the target female population and significantly increase vaccination for boys by 2030 [5]. Currently, 30 European countries have introduced HPV vaccination into their national immunization programs recommending vaccination for preadolescent females, and 17 countries for males [6]. In Italy, vaccination has been offered actively and free of charge to girls in the twelfth year of life since 2007. Afterward, the active offer was gradually extended to older girls, and since 2015 some administrative regions, including Sicily, have begun to offer the vaccine to boys as well [7]. Despite the proven efficacy and safety profile, vaccination coverage for HPV is still unsatisfactory in several European countries; for example, in Italy, the achieved coverage of HPV vaccination was 47% in 2015, far from the desirable 95% coverage [8,9]. The HPV vaccination coverage suffered a further decline in 2020 due to difficulties and obstacles related to the COVID-19 pandemic; considering the whole Italian territory, vaccination coverage for HPV was 30.3% for girls born in 2008. Even in Sicily the vaccination rate dropped drastically, with values of 34.6%, 35.9%, and 22.6% for females and 26.9%, 24%, and 14.9% for males in the 2006, 2007, and 2008 birth cohorts, respectively [10].

The most frequent reasons reported in the literature for vaccination refusal were fear and concerns about adverse effects, safety and newness, poor perception of risk related to infection and subsequent diseases, lack of perception of direct clinical value (especially for males), and lack of accurate information [11,12]. There are several cognitive elements that can impact vaccination adherence. The cognitive models can explore people’s perceptions about benefits of prevention and propensity to adhere to the preventive practices, one example is the Health Belief Model (HBM) [13]. Developed in the 1950s, the HBM continues to be one of the most accredited and widely used theories to investigate people’s perceptions about the benefits of prevention and, at the same time, obstacles associated with adhering to preventive practices, thus allowing the model to predict the behavior adopted, such as the decision to get vaccinated [13,14]. Another element that can influence vaccination adherence is vaccination literacy. It originates from health literacy (HL), and identifies the degree to which people have the capacity to obtain, process, and understand basic information and services to make appropriate health decisions [15,16]. Poor HL is associated with a lower use of preventive services and with a reduced adoption of protective behaviors such as immunization [17]. There are numerous tools used to measure HL, such as short questionnaires and tests suitable for evaluating multiple skills: the Test of Functional Health Literacy in Adults (TOFHLA), which measures reading fluency; the Rapid Estimate of Adult Literacy in Medicine (REALM) test, which evaluates pronunciation and vocabulary domain; the rapid test of Newest Vital Sign (NVS) screening on the understanding of nutritional information; the Health Activities Literacy Scale (HALS) with prose elements; the Single Item Literacy Screener (SILS) test, which consists of a single question; and the Medical Term Recognition Test (METER), which consists of a list of 70 medical and nonmedical terms [18,19,20].

The main aim of this study was to identify the determinant factors for HPV vaccination acceptance by exploring the cognitive models and HL level of adolescent and young adults in a low vaccination coverage setting.

## 2. Materials and Methods

A cross-sectional study was conducted among students and young adults, including males and females (13–26 years old). The survey was conducted through the administration of a questionnaire to young adults attending the University of Palermo and young students attending high schools in the Palermo area from April to December 2019. The questionnaire consisted of 4 different sections. The first one concerned sociodemographic information, with questions about age, sex, family, mother’s and father’s educational levels, and economic status. The second one included questions on exposure to risk factors and personal behaviors such as smoking, drinking alcohol, physical activities, dietary habits, and sexual behaviors. The third section focused on knowledge about HPV infection and vaccination. In this section, the questions also explored compliance with the vaccination schedule based on the number of doses received, which allowed the identification of vaccinated and unvaccinated students. The fourth section consisted of the HBM section, which investigated, through 15 items, the six different domains of the model and, in this case, the benefits related to general vaccinations and HPV vaccination, barriers to general vaccination and the HPV vaccine, susceptibility to HPV infection, and severity of diseases related to HPV infection. The available response options, according to a Likert scale, ranged from 1 = absolutely not to 5 = very highly. The HMB domain score was then recategorized according to the following value: 0 = 1(”I strongly disagree”) and/or 2 (”I disagree”) or 1 = 3 (”Neither I agree nor I disagree”), 4 (”I agree”), and/or 5 (”I strongly agree”). The recategorized score was used for all questions, and the cut-off was selected by using the median value for “High HBM” (score ≥ 12), which reflected a greater awareness of the importance of vaccination; and “Medium-Low” HBM (score < 12), which reflected a lower tendency to adhere to vaccination. In this section, we also included two self-administering tests for the evaluation of HL. The first was the SILS test, consisting of a single question (i.e., How often do you need to have someone helping you when you read instructions, pamphlets or other written material from your doctor or pharmacy?) with five answer options (1 = never; 2 = rarely; 3 = sometimes; 4 = often; 5 = always) [20]. The second was the METER test, which consists of a list with 70 terms, including 40 medical and 30 nonmedical terms printed on a single page; the score is assigned based on the number of identified medical words. According to the score, it was possible to identify three levels of HL as follows: 0–20 = low level; 21–34 = marginal level; 35–40 = adequate level [19].

## 3. Statistical Analysis

The normality of the distribution for the quantitative variables was assessed with the skewness and kurtosis test. The absolute and relative frequencies were calculated for the qualitative variables. The association of quantitative variables normally and not normally distributed with adhesion to the HPV vaccination were evaluated respectively with the Student’s *t*-test and with the Wilcoxon and Mann–Whitney test, while for the qualitative variables, the chi-squared test was used. A univariable logistic regression analysis was performed to evaluate the factors associated with acceptance of HPV vaccination. Study covariates that were found to be significantly associated with adherence to the vaccination after the univariable analysis (*p* < 0.05) were evaluated in a multivariable logistic regression model. All collected data were analyzed using Stata MP v14.2 statistical software.

## 4. Results

Overall, 3073 students were enrolled, of which 2081 (67.7%) were university students (14.7% attending medical courses) and 992 (32.3%) were high school students. The majority of people interviewed were female (71.1%) aged 21–23 years (26.1%). The parents of the interviewed students were more frequently workers (father 99% and mother 85.6%), and the economic status was more frequently defined as good (47.9%) or acceptable (44.4%). As for lifestyle habits, 22% of the people enrolled reported to be smokers, and 71.5% to drink alcohol occasionally. With regard to sexual habits, 48.5% of participants were in an exclusive relationship at the time of the survey, 91.1% were heterosexual, and 43% had their sexual debut between the ages of 15 and 18. More than half said they used condoms regularly during sexual intercourse, while 22.3% did not use any contraceptive method. The analysis also made a comparison between vaccinated and unvaccinated students. In details, vaccinated students were more frequently female (88.3% vs. 63.7%, *p* < 0.001), attending high school (38.4% vs. 29.7%, *p* < 0.001), and 13–14 years old (10.4% vs. 4.7%, *p* < 0.001). It was also found that those who adhered more frequently to vaccination were students with working mothers (88.9% vs. 84.2%, *p* < 0.001) with a good economic condition (53.5% vs. 45.4%, *p* < 0.001) and bisexual students (6.2% vs. 3.9%, *p* < 0.028) (Table 1).

Most of the respondents reported to be informed about sexually transmitted infections (STIs) (87.8%), but only one-third (34.9%) recognized HIV as the most frequent sexually transmitted agent, followed by genital herpes (28.2%). The majority of students had a high level of knowledge of HPV infection and vaccination: the highest level of knowledge was on HPV transmission (91.8%), followed by association between HPV and cervical cancer (65.3%), but many of them believed that efficacious therapies are available (80.1%). The main source of information on HPV vaccination was school (21%), followed by parents (17.3%) and pediatricians or general practitioners (16.8%). Less than a third of the students said they had been immunized against HPV and completed the vaccination cycle (*n* = 925, 30.1%), as reported in Table 2. Furthermore, among those unvaccinated, there were students more frequently informed about the vaccine at school (16.7% vs. 22.9%, *p* < 0.001), through the internet (6.7% vs. 14.5%, *p* < 0.001), by media (4.4% vs. 10.2%, *p* < 0.001), or by friends (0.8% vs. 5.1%, *p* < 0.001) (Table 2).

With regard to HL, the SILS test results showed that the majority of respondents believed they did not need any help when reading or understanding medical indications (43.6% “Rarely” and 31.6% “Never”). When evaluating the HL level with the METER test, 52.1% (*n* = 1610) of them had a good score, identifying 35–40 correct terms. The results obtained regarding the HBM investigation showed that 35.7% of the students had a high level of HBM, with a score greater than or equal to 12. Furthermore, vaccinated in comparison to unvaccinated students more frequently answered “Rarely” to the SILS test (47.1% vs. 42%, *p* = 0.005) and had a higher HBM score (56.8% vs. 26.7%, *p* < 0.001) (Table 3).

Considering the sample of students interviewed, the factors directly associated with HPV vaccination adherence in the multivariable analysis were female sex (OR = 4.43, *p* < 0.001), high HBM total score (OR = 4.23, *p* < 0.001), good HL level, and “Rarely” response in the SILS test (OR = 1.26, *p* = 0.047). Furthermore, having parents (OR = 1.78, *p* = 0.004), general practitioners (OR = 1.88, *p* = 0.001), and educational material provided by public vaccination services (OR = 1.97, *p* = 0.001) as HPV vaccine information sources were also directly associated with being vaccinated. On the other hand, factors inversely associated with vaccination adherence were: being 17–18 years old (OR = 0.19, *p* < 0.001); and HPV vaccine information sources such as school (OR = 0.67, *p* = 0.05), internet (OR = 0.62, *p* = 0.043), and friends (OR = 0.28, *p* = 0.007) (Table 4).

## 5. Discussion

The study revealed low vaccination HPV coverage among adolescents and young adults in the Palermo area. Although the HPV vaccine is effective in reducing the burden of HPV-related diseases, the vaccine’s acceptability should be increased to achieve the desirable 95% vaccination coverage [21].

In this study, a factor associated with vaccination adherence was the female sex. The difference in vaccine acceptability was already highlighted in similar studies. A few studies showed that females had a higher knowledge of HPV infection and prevention strategies, such as screening, that could increase the probability of being vaccinated [22,23]. An approach to increasing adherence in males might be to focus on the role of the vaccine in protecting against HPV infections. This message should oppose the general idea that the HPV vaccine aims at exclusively preventing cervical cancer, which affects only females [24].

This survey found that the probability of being vaccinated could vary according to the different age groups. In particular, the age cohort between 17 and 18 years old was less vaccinated. An explanation of this finding could be that this age might correspond to sexual debut, exposing young people to a high risk of HPV infection [25]. Furthermore, adolescents at this age do not talk about health issues with their family for several reasons: limited understanding of the infection, uncertainty about the need for vaccination, and embarrassment in discussing sexual habits with parents [26]. Consequently, health-promotion campaigns are needed to encourage parents to talk with their children about their sexual habits and the related medical topics.

Another factor directly associated with HPV vaccination was a high HBM score. The influence of behavioral characteristics was already explored in other studies. For example, in Sweden, girls with a higher HBM score in HPV disease susceptibility had a higher intention to adopt preventive strategies [27]. On the other hand, a previous study conducted in Sicily showed a higher influence of HPV vaccinations’ perceived benefits in preventive strategy acceptance [28]. According to these findings, an in-depth analysis of the behavioral model using a theoretical framework such as HBM should be performed before selecting the intervention to increase HPV vaccine adherence.

Moreover, informative sources on HPV vaccination can play a major role in the decision-making process. For example, among the sources directly associated with HPV vaccination, there were general practitioners, public health services and parents. The associations between these sources of information and vaccination acceptability was confirmed by other studies [29,30]. Furthermore, an exponential effect could be reached when several informative sources from the medical perspective used a common shared message for young adults.

On the other hand, there were sources inversely associated with HPV vaccination adherence: the internet, school and friends. The role of the internet in providing information on the HPV vaccine is particularly ambiguous because there are many websites that deal with the topic, but some of these are managed by people who are not competent in the medical field [31]. For this reason, it must be considered very useful to teach young people to seek information on medical topics only on authorized and reliable websites. For example, in 2000, the European Union founded the MedCERTAIN certification system, which has rating-based content to inform users on the quality of websites [32]. Another tool used to increase reliability of online sources is the HONcode, a certified logo that testifies to the truthfulness of the sources and to the author’s credentials [33].

Another source of information inversely associated with HPV vaccination adherence was the school. A lack of medical information can be the cause of a changing role of schools as informative sources. Indeed, in 2004, the Italian law relating to school medical service was repealed, as it was dated and referred to a very different epidemiological context. Since then, the school health service has been marginally managed through the responsibility of local pediatricians [34]. This theory is in accordance with other studies showing that the active presence of health professionals in a school setting can greatly improve students’ acceptability of the HPV vaccine. For example, in Sweden, face-to-face interviews between nurses and students led to improvements in beliefs about HPV prevention; while in Norway, there was a great increase in vaccination coverage after the start of the school-based HPV vaccination program [35,36].

Moreover, friends were revealed as a source of information that was inversely associated with HPV vaccination. In the literature, friends had a mixed role as informative sources. For example, a study conducted in the U.S.A. identified friends among the most common sources of information, highlighting how the decision-making process for HPV vaccination was influenced by the contexts of daily life [37]. On the other hand, another U.S.A. survey aimed at guiding future interventions to target peer influence on medical decisions, and it found that friends negatively affected young adults’ knowledge about HPV [38]. The different levels of education, as well as the social and cultural context, could be responsible for these conflicting results [39]. It should be noted that the people interviewed had a low level of knowledge about HPV diseases or the HPV vaccine, and they could also have friends with a low level of knowledge. Consequently, it would be advisable for young people to be informed about HPV vaccination by more accredited and competent sources, rather than by word of mouth among friends.

A final consideration should be made on HL. Indeed, the analysis showed that students with a good level of HL had a higher probability of being vaccinated against HPV. The higher vaccine acceptance could be related to the linking of HL with higher knowledge of both the virus and available vaccines [40]. This study was the first to our knowledge to demonstrate an association between a high level of HL and an effective adherence to HPV vaccination. This evidence can allow the building of preventive strategies based on the Sicilian HL level. Indeed, there are several countries that have implemented HL improvement policies, and that understood how it can represent an important tool for public health. For example, in Australia, the “National Declaration on Health Literacy” was published in 2014, outlining three areas of intervention: (i) embedding health literacy into systems; (ii) ensuring effective communication; and (iii) integrating health literacy into education [41].

The main limitation of this study was the recall bias due to the delay between HPV vaccination practice and questionnaire administration. Furthermore, data on HPV vaccination schedule were self-reported by those interviewed to ensure anonymization of the questionnaires, and therefore may have been subject to biases and limitations. Notwithstanding previous limitations, this study explored the association among HPV vaccination acceptance, literacy level, and behavioral motivation after 15 years of vaccine implementation in a population with low vaccination coverage.

## 6. Conclusions

HPV vaccination acceptance is directly influenced by the level of HL, the perception of severity, susceptibility to disease, vaccination barriers, and benefits. This evidence should be used to build interventions that improve HPV vaccine acceptability. However, the role of a shared message by the strongly associated informative sources and the implementation of health information campaigns in weak informative settings, such as in schools, also can allow for achieving desirable vaccination coverage rates.

## Figures and Tables

**Table 1 vaccines-10-00998-t001:** Socio-demographic and behavioral characteristic of enrolled students and differences between fully or not vaccinated for HPV.

		Total Respondents	Vaccinated *n* (%)	Unvaccinated *n* (%)	*p*-Value
		3073	925 (30.1%)	2148 (69.9%)	
Age	13–14 years old	199 (6.5%)	97 (10.5%)	102 (4.8%)	<0.001
	15–16 years old	419 (13.6%)	185 (20.0%)	234 (10.9%)	
	17–18 years old	368 (11.9%)	70 (7.6%)	298 (13.9%)	
	19–20 years old	500 (16.3%)	204 (22.1%)	296 (13.8%)	
	21–23 years old	801 (26.1%)	265 (28.7%)	536 (24.9%)	
	24–26 years old	539 (17.5%)	801 (8.6%)	459 (21.4%)	
	>26 years old	247 (8.0%)	24 (2.6%)	223 (10.4%)	
Sex	Male	888 (28.9%)	108 (11.7%)		<0.001
				780 (36.3%)	
	Female	2185 (71.1%)	817 (88.3%)	1368 (63.7%)	
Attended school/education	Medicine faculty	453 (14.7%)	570 (61.6%)	1511 (70.3%)	<0.001
	Other faculties	1628 (53%)			
	Lyceum	876 (28.5%)	355 (38.4%)	637 (29.7%)	
	Technical institutes	116 (3.8%)			
Father	Primary school diploma	763 (25.50%)	226 (24.9%)	537 (25.8%)	0.623
	High school diploma	1321 (44.2%)	395 (43.6%)	926 (44.5%)	
	Graduation	903 (30.2%)	285 (31.5%)	618 (29.7%)	
	Worker	3043 (99%)	920 (99.5%)	2123 (98.8%)	0.107
	Health worker	273 (8.9%)	84 (9.2%)	189 (8.8%)	0.768
Mother	Primary school diploma	687 (23%)	180 (19.8%)	507 (24.3%)	0.026
	High school diploma	1396 (46.6%)	437 (48.1%)	959 (46.0%)	
	Graduation	910 (30.4%)	291 (32.1%)	619 (29.7%)	
	Worker	2631 (85.6%)	822 (88.9%)	1809 (84.2%)	0.001
	Health worker	264 (8.8%)	74 (8.2%)	190 (9.1%)	0.414
Economic status	Good	1472 (47.9%)	495 (53.5%)	977 (45.4%)	<0.001
	Acceptable	1363 (44.4%)	374 (40.4%)	989 (46.0%)	
	Low	238 (7.7%)	56 (6.1%)	182 (8.5%)	
Smoking		676 (22%)	177 (19.1%)	499 (23.2%)	0.012
Alcohol	Never	541 (17.6%)	191 (20.7%)	350 (16.3%)	<0.001
	occasionally	2196 (71.5%)	655 (70.8%)	1541 (71.7%)	
	2 or 3 times in a week	312 (10.8%)	68 (7.4%)	244 (11.4%)	
	every day	24 (0.8%)	11 (1.2%)	13 (0.6%)	
Sexual orientation	Heterosexual	2081 (91.1%)	836 (90.4%)	1965 (91.5%)	0.028
	Homosexual	87 (2.8%)	20 (2.2%)	67 (3.1%)	
	Bisexual	142 (4.6%)	57 (6.2%)	85 (3.9%)	
Relationship status	Single	1357 (44.2%)	415 (44.9%)	942 (43.9%)	0.872
	Exclusive relationship	1491 (48.5%)	439 (47.5%)	1052 (49%)	
	Nonexclusive relationship	157 (5.1%)	50 (5.4%)	107 (5%)	
First sexual intercourse	12–14 years old	203 (6.6%)	62 (6.7%)	141 (6.6%)	<0.001
	15–18 years old	1321 (43%)	373 (40.6%)	945(44.9%)	
	<18 years old	619 (20.1%)	157 (16.9%)	462 (21.5%)	
	Condom	1735 (56.5%)	487 (52.7%)	124 (58.1%)	<0.001
Contraceptive method used	Oral contraceptive	276 (9%)	104 (11.2%)	172 (8.0%)	0.004
	Nothing	696 (22.6%)	211 (22.8%)	485 (22.6%)	0.888

**Table 2 vaccines-10-00998-t002:** HPV infection and HPV vaccine knowledge, and differences between fully or not vaccinated for HPV.

		Total Respondents	Vaccinated *n* (%)	Unvaccinated *n* (%)	*p*-Value
		3073	925 (30.1%)	2148 (69.9%)	
Have you been informed about sexually transmitted infections?	Yes	2699 (87.8%)	820 (88.6%)	1879 (87.5%)	0.362
	Papilloma virus	568 (18.5%)	160 (17.3%)	409 (19.1%)	0.254
What is the most common sexually transmitted disease?	HIV	1074 (34.9%)	337 (36.4%)	737 (34.3%)	0.258
	Genital herpes	867 (28.2%)	276 (29.8%)	591 (27.5%)	0.189
Do you think that HPV infection is sexually transmitted?	Yes	2821 (91.8%)	834 (90.2%)	1987 (92.5%)	0.03
Do you think that the cervical cancer is due to HPV?	Yes	2008 (65.3%)	646 (69.8%)	1362 (63.4%)	0.001
Do you think that the penile cancer is due to HPV?	Yes	559 (18.2%)	176 (19.0%)	383 (17.8%)	0.43
Do you think that the prostate cancer is due to HPV?	Yes	609 (19.8%)	208 (22.5%)	401 (18.7%)	0.015
Do you think that the Herpes infection is due to HPV?	Yes	282 (7.1%)	68 (7.4%)	214 (9.9%)	0.021
Do you think HPV can cause infertility?	Yes	282 (9.2%)	77 (8.32%)	142 (6.6%)	0.09
Are there therapies for HPV infection?	Yes	2460 (80.1%)	745 (80.5%)	1715 (79.8%)	0.657
Have you ever heard about HPV vaccine?	Yes	2672 (87%)	882 (95.4%)	1790 (83.3%)	<0.001
	Parents	532 (17.3%)	257(27.8%)	275 (12.8%)	<0.001
	School	646 (21%)	155 (16.7%)	491 (22.9%)	<0.001
	Internet	374 (12.2%)	62 (6.7%)	312 (14.5%)	<0.001
What was your main source of information about the HPV vaccine?	Information material given by public vaccination services	134 (4.4%)	58 (6.3%)	76 (3.5%)	0.001
	Media	259 (8.4%)	41 (4.4%)	218 (10.2%)	<0.001
	General pratictioner or pediatrician	515 (16.8%)	241 (26.1%)	274 (12.8%)	<0.001
	Friends	116 (3.8%)	7 (0.8%)	109 (5.1%)	<0.001

**Table 3 vaccines-10-00998-t003:** HBM and HL scores, and differences between fully or not vaccinated for HPV.

		Total Respondents	Vaccinated *n* (%)	Unvaccinated *n* (%)	*p*-Value
		3073	925 (30.1%)	2148 (69.9%)	
HBM (Total score = 15)	High HBM (≥12)	1098 (35.7%)	525 (56.8%)	573 (26.7%)	<0.001
	Low HBM (<12)	1975 (64.3%)	400 (43.2%)	1575 (73.3%)	
	Never	971 (31.6%)	247(26.7%)	724 (33.7%)	0.005
	Rarely	1339 (43.6%)	436 (47.1%)	903 (42.0%)	
SILS test	Sometimes	569 (18.5%)	183 (19.8%)	386 (17.9%)	
	Often	158 (5.1%)	48 (5.2%)	110 (5.1%)	
	Always	36 (1.2%)	11 (1.2%)	25 (1.2%)	
	Low level (0–20)	596 (19.4%)	163 (17.6%)	433(20.2%)	0.096
METER test	Marginal level (21–34)	876 (28.5%)	285 (30.8%)	591(27.5%)	
	Adequate level (35–40)	1601 (52.1%)	477 (51.6%)	1124 (52.3%)	

**Table 4 vaccines-10-00998-t004:** Univariate and multivariate analysis of factors associated with HPV vaccination acceptance.

		Crude OR	*p*-Value	Adjusted OR	*p*-Value
Students	Palermo High School	Ref			
	Palermo University	0.67	<0.001	0.54	0.57
Sex	Female vs. Male	4.31	<0.001	4.43	<0.001
Age	13–14 years old	Ref			
	15–16 years old	0.83	0.285	1.06	0.778
	17–18 years old	0.25	<0.001	0.19	<0.001
	19–20 years old	0.72	0.056	0.96	0.969
	21–23 years old	0.52	<0.001	0.7	0.747
	24–26 years old	0.18	<0.001	0.23	0.181
	>26 years old	0.11	<0.001	0.17	0.121
Mother	Worker vs. nonworker	1.49	0.001	1.37	0.064
Mother	Educational qualification	1.32	0.012	0.88	0.372
	Graduation				
Economic status	Low	Ref			
	Acceptable	1.23	0.21	0.93	0.735
	Good	1.64	0.002	1.08	0.708
Smokers	Yes vs. no	0.78	0.012	0.88	0.312
Sexual orientation	Heterosexual	Ref			
	Homosexual	0.7	0.17	1.16	0.646
	Bisexual	1.57	0.01	1.14	0.548
SILS test	Always	1.29	0.491	0.87	0.762
	Often	1.28	0.191	0.76	0.252
	Sometimes	1.39	0.005	1.03	0.807
	Rarely	1.42	<0.001	1.26	0.047
	Never	Ref			
Meter test	Low level	Ref			
	Marginal level	1.28	0.034	0.89	0.422
	Adequate level	1.13	0.262	0.87	0.323
High HBM	Total score ≥ 12 vs. <12	4.84	<0.001	4.23	<0.001
Believing that HPV causes cervical cancer	Yes vs. no	1.34	0.001	0.96	0.743
Believing that HPV causes Herpes infection	Yes vs. no	0.72	0.022	0.67	0.044
Believing that HPV causes prostate cancer	Yes vs. no	1.26	0.015	0.9	0.489
Having heard about HPV vaccination	Yes vs. no	4.1	<0.001		
Information source about HPV vaccine	Parents (yes vs. no)	2.62	<0.001	1.78	0.004
	General pratictioner or paediatrician (yes vs. no)	2.41	<0.001	1.88	0.001
	Public vaccination services (yes vs. no)	1.82	0.001	1.97	0.001
	Teachers/school (yes vs. no)	0.68	<0.001	0.67	0.05
	Internet (yes vs. no)	0.42	<0.001	0.62	0.043
	Media (yes vs. no)	0.41	<0.001	0.62	0.067
	Friends (yes vs. no)	0.14	<0.001	0.28	0.007

## Data Availability

Data will be made available by the corresponding author upon reasonable request.

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
