# Peer review of "Acceptability of HPV Vaccination in Young Students by Exploring Health Belief Model and Health Literacy"

_vaccines, 2022, doi:10.3390/vaccines10070998_

Round 1

Reviewer 1 Report

I read the article entitled "Acceptability of HPV vaccination in young students exploring Health Belief Model and Health Literacy”. It is an interesting and original article.  I would recommend this paper to be published after minor corrections.

  Introduction: I would suggest that the authors increase the references especially on the reasons for refusing to vaccinate (lines 57-59).

Results: Table 1 is particularly broad I would suggest the authors to split it into more tables.

Author Response

I read the article entitled "Acceptability of HPV vaccination in young students exploring Health Belief Model and Health Literacy”. It is an interesting and original article.  I would recommend this paper to be published after minor corrections.

  Introduction: I would suggest that the authors increase the references especially on the reasons for refusing to vaccinate (lines 57-59).

Answer:

Thanks for your suggestion, reporting accurately the most common reasons for rejection of the HPV vaccine is very important in the context of the article and we have explained it, expanding the references, at lines 59 - 62.

Results: Table 1 is particularly broad I would suggest the authors to split it into more tables.

Answer

Your suggestion was gladly accepted. We thought it appropriate to split Table 1 into several parts and this allowed us to broaden the description of the results.

Reviewer 2 Report

Falluca et.al., propose to study the acceptability of HPV vaccination in young students in Italy. And it seems that it concludes that only 30% of participants received all three doses and the authors recommend improvements in programs focused on increasing health literacy are needed.

Overall, the paper is poorly written and should be edited for context.  I had to guess as to what was been stated/said.  It seems that the study is of only 12 year olds, but not clear, thus difficult to assess the quality of the work. I do think that studying reason for low adherence to the vaccine schedule is important.

Comments:

Title: are the students exploring Health Belief model? Or was the study performed using the Health belief model? Is there a Health Literacy model too?

Abstract makes it sound that vaccine was only offered free to 12 year olds.  Do authors mean 12 year old and older? Starting at 12 years of age until 26? Or is it really 12 year olds only. Or is the study looking at only 12 year olds?  The data reported on the abstract…is that for the 12 year olds? Or all the vaccinations in Italy?

Line 19, why is Health Literacy capitalized? What is Health Literacy.

Lines 21-24, are the numbers significant? Just listing them with no explanation/context is not useful. (likert scale?)

Line 26, what is the desired vaccination coverage? (should be in the abstract)
Key words: there is also a behavioral model?

Body:

Lines 79-80 make it sound that you studied a population known to have lower vaccination acceptance.  Is this correct? Did you know the vaccination values were low prior to the study?

Line 83-85, 13 year olds are in High School?

Line 88 Exposure not exposition

HBM questions make sense and seemed to have influenced the analysis.

Description of the results need be expanded.  Lots of data presented, but what influence some of it, is unclear.  There is data that was collected, but not sure how it related to the study.

The body was better written, but clear conclusions are difficult to find. 

Did you use any of the socioeconomic data?

Minor comments:

Line 12 Twelve, not twelfth

Line 70 associate with, not associated to

Overall: the idea/need to understand why there is low compliance of the vaccine is important.  The abstract needs major re-writing.  There is a lot of data, some not even discussed, thus this lack of focus on what is important diminishes the impact of the study.  I think that the data is collected, but lacks a thorough evaluation and needs to be better separated/described.  I had a difficult time following the writing because it didn’t flow.  Reading the conclusion…it doesn’t include many findings that seem important, for example a working mother, or sexual habits (bisexual), or economic status. This is not mentioned in discussion. 

Author Response

Fallucca et.al., propose to study the acceptability of HPV vaccination in young students in Italy. And it seems that it concludes that only 30% of participants received all three doses and the authors recommend improvements in programs focused on increasing health literacy are needed.

Overall, the paper is poorly written and should be edited for context. I had to guess as to what was been stated/said. It seems that the study is of only 12 year olds, but not clear, thus difficult to assess the quality of the work. I do think that studying reason for low adherence to the vaccine schedule is important.

Comments:

Title: are the students exploring Health Belief model? Or was the study performed using the Health belief model? Is there a Health Literacy model too?

Answer

The study was conducted on a large population between the ages of 13 and 26; university and high school students were interviewed to investigate the factors influencing the acceptability of HPV vaccination (lines 82- 85, Table 1). The study was conducted using the Health Belief Model, which is one of the most accredited models for investigating people's perceptions of the benefits of prevention, including vaccination practice (lines 61 – 66, lines 92 – 103). It is not entirely correct to speak about "Health literacy model" but there are several tests used to investigate the level of Health Literacy, as poor level is associated with a lower propensity for prevention (lines 66 – 77). We modified the title in “Acceptability of HPV vaccination in young students by exploring Health Belief Model and Health Literacy” to make it more readable.

Abstract makes it sound that vaccine was only offered free to 12 year olds. Do authors mean 12 year old and older? Starting at 12 years of age until 26? Or is it really 12 year olds only. Or is the study looking at only 12 year olds?  The data reported on the abstract is that for the 12 year olds? Or all the vaccinations in Italy?

Answer

Thank you for your suggestion. It is certainly appropriate to specify in the abstract that in Italy the vaccine was initially offered free of charge only to the primary target (twelfth year of life) but, over time, the offer has been expanded. This study, in fact, looking at a large population; university and high school students were interviewed as specified in the abstract “For more than 15 years, in Italy, HPV vaccine has been offered free of charge to girls from the age of 12. Over time, the free offer of the HPV vaccine has also been extended to boys and to young adults at risk of developing HPV lesions”.

Line 19, why is Health Literacy capitalized? What is Health Literacy.

Answer

Health literacy is a multifaceted concept that deals with the capacities of people to meet the complex demands of health in a modern society. It was only capitalized to avoid length of the text. HL identifies the degree to which people have the capacity to obtain, process and understand basic information and services to make appropriate health decisions (lines 71 – 73). Poor HL is associated with to a lower use of preventive services and with a reduced adoption of protective behaviours such as immunization.

The relationship between HL and vaccination (including vaccine aptitude and vaccine uptake) has already been explored in the literature and it is believed that a low level of HL is associated with less recourse to prevention services and reduced adoption of protective behaviours such as immunization (reference 15 "Ratzan C.S."; reference 16 "Lorini C. et. al.")

Lines 21-24, are the numbers significant? Just listing them with no explanation/context is not useful. (likert scale?)

Answer

For each factor listed in lines 21 - 24 the P-value is reported which, if less than 0.05, testifies to the significance of the result. The results of the multivariable analysis significantly associated with the acceptance of the HPV vaccination were more deeply discussed in the "Discussion" section but not in the abstract section in order to avoid extremely length.

Line 26, what is the desired vaccination coverage? (should be in the abstract)

Answer

Thanks for the suggested correction. The value of the anti-HPV vaccination coverage is reported in the introduction and it is reported the abstract section “Further health promotion programs focused on improving health literacy and perception of the HPV vaccine benefits should be implemented in order to achieve the desirable 95% vaccination coverage”.

Key words: there is also a behavioural model?

Answer

The questionnaire adopted a Health Belief Model that is an accredited behavioural model explaining the adoption of health behaviours. It has been used extensively to study vaccine beliefs and behaviours, and has also been used in vaccination research to identify people's perceptions of disease and vaccination. Furthermore, an entire section of the questionnaire was devoted to the survey on the "personal behaviour" and "lifestyle" of the interviewed students (lines 90 - 92).

Body:

Lines 79-80 make it sound that you studied a population known to have lower vaccination acceptance. Is this correct? Did you know the vaccination values were low prior to the study?

Answer

As reported in the abstract, vaccination coverage for HPV in Italy is quite low and far from the desired 95% rate, especially in some regions such as Sicily. “Although HPV vaccine effectiveness and availability, vaccination coverage is low in Italy with a reported value of 46.5% in 2020. Furthermore, in Southern administrative Regions, vaccination coverage is even lower than national values, with 25.9% coverage in Sicily.” This study analyse the vaccination acceptance in a sample of people with low HPV vaccination coverage.

Line 83-85, 13 year olds are in High School?

Answer

The students interviewed were directly asked to indicate their age. It is very common in Italy to start high school at 13 and turn 14 in the first year of high school.

Line 88 Exposure not exposition

Answer

Thanks for the suggested correction, it was well received. “The second one regarding exposure to risk factors and personal behaviours, with questions about smoking, alcohol, physical activities, diet food and sexual behaviours” (lines 91- 92).

HBM questions make sense and seemed to have influenced the analysis.

Answer

The description of the domains studied to explore the Health Belief Model is presented at lines 94 - 98. We used 15 questions to assess the perception of the benefits and barriers of general vaccination and HPV vaccination in detail, as well as the perception of susceptibility and severity of infection and HPV related diseases. A score was assigned to each single item of the 15 proposed based on the response of the students interviewed. The overall score for HBM was assessed by identifying people with a "high level of total HBM" and people with a "medium / low level" (lines 98 - 104).

The analysis showed that students with a high level of HBM (correct perception of the benefits of vaccination and the severity of the related HPV disease) were 4 times more likely to be fully vaccinated against HPV (OR= 4.23; p-value <0.001) (line 165, Tab 4).

Description of the results need be expanded.  Lots of data presented, but what influence some of it, is unclear.  There is data that was collected, but not sure how it related to the study.

Answer

We considered it appropriate to split table 1 to allow a better understanding of the results and to expand the description of the findings.

The present study was carried out with the intent to evaluate which factors significantly influenced the acceptance or rejection of HPV vaccination in a population of students aged 13 to 26 years. Many factors were investigated: personal and socio-demographic characteristics, personal behaviours, lifestyle, knowledge of HPV infection and the HPV vaccine, sources of information, HBM, HL etc.).

The logistic regression analysis showed what factors are actually associated with adhering to vaccination. The description of the results and the arguments presented in the "Discussion" paragraph have been focused on the significant findings. However, in the tables, the values relating to all the investigated factors have been faithfully reported.

The body was better written, but clear conclusions are difficult to find. 

Answer

In the “Conclusion” paragraph we stressed, once again, which factors were directly associated with HPV vaccination in the analysis; it could be very useful to act on these factors to improve vaccination coverage against HPV (for example, it would be appropriate to improve the level of health literacy or to invest in appropriate sources of information).

Did you use any of the socioeconomic data?

Answer

It is certainly important to evaluate the influence of the socio-economic level in determining the acceptance of vaccination and in our study the economic status and social-family context were investigated with questions related to the occupation and educational qualification of the parents. However, multivariate logistic regression did not show these factors as significantly associated (Tab 4).

Minor comments:

Line 12 Twelve, not twelfth

Line 70 associate with, not associated to

Answer

Thanks for the suggested corrections. “Poor HL is associated with a lower use of preventive services and with a reduced adoption of protective behaviours such as immunization” (line 73).

Overall: the idea/need to understand why there is low compliance of the vaccine is important.  The abstract needs major re-writing.  There is a lot of data, some not even discussed, thus this lack of focus on what is important diminishes the impact of the study.  I think that the data is collected, but lacks a thorough evaluation and needs to be better separated/described.  I had a difficult time following the writing because it didn’t flow.  Reading the conclusion…it doesn’t include many findings that seem important, for example a working mother, or sexual habits (bisexual), or economic status. This is not mentioned in discussion. 

Answer

The abstract was implemented according to suggestion of reviewer and the data description were split to be more accurate. The conclusion were supported by the main results of the analysis that should be applicable in the population of Sicilian adolescent for which can be representative. Finally, the language was checked by an English native speaker.

Reviewer 3 Report

I would suggest comparing the HPV vaccination adherence with the country vaccination policy and the entire country acceptance of HPV vaccine.

Author Response

I would suggest comparing the HPV vaccination adherence with the country vaccination policy and the entire country acceptance of HPV vaccine.

Answer

Thank you very much for the suggestion. Italian vaccination offer policy and HPV vaccine acceptance levels have been explained at lines 48 - 58.

Round 2

Reviewer 2 Report

Thanks to the authors for their thorough review/editing of the manuscript.  It is an important study that will inform the local and national community regarding the need of improved vaccination efforts.